# Soybean Response to N Fertilization Compared with Co-Inoculation of *Bradyrhizobium japonicum* and *Azospirillum brasilense*

Jose Bais [1],*, Hans Kandel [1], Thomas DeSutter [2], Edward Deckard [1] and Clair Keene [1]

[1] Department of Plant Sciences, North Dakota State University, Fargo, ND 58108, USA; hans.kandel@ndsu.edu (H.K.); edward.deckard@ndsu.edu (E.D.); clair.keene@ndsu.edu (C.K.)
[2] School of Natural Resource Sciences, North Dakota State University, Fargo, ND 58108, USA; thomas.desutter@ndsu.edu
* Correspondence: jose.bais@ndsu.edu

**Abstract:** The soybean [*Glycine max* (L.) Merrill] relationship with the bacteria *Bradyrhizobium japonicum* is responsible for providing around 60% of the nitrogen (N) required for the crop and the remaining N comes from the soil or supplemental fertilization. To investigate if higher yields are possible, supplemental N studies and co-inoculation of *Rhizobium* with *Azospirillum* are necessary. This N rate (0, 30, 56, 112, 336 kg N ha$^{-1}$) and inoculation study was conducted across eight environments in eastern North Dakota, USA, in 2021 and 2022. Also, the effect of supplemental N and co-inoculation on nodulation was evaluated. When N was applied at 112 kg N ha$^{-1}$, nodulation was significantly inhibited. Co-inoculation increased the number of large nodules and the volume of nodules; however, the yield was not different from inoculation with *B. japonicum*. Nitrogen at 112 and 336 kg ha$^{-1}$ increased grain yield, protein yield, and seed weight; however, the higher N rate decreased plant population. There were significant positive relationships between yield and protein content and seed weight, and negative relationships between oil and protein content, and yield and oil content. Based on a polynomial relationship, the highest yield (3711 kg ha$^{-1}$) would be achieved at 273 kg N ha$^{-1}$. The application of N resulted in a yield increase but using current prices may not be an economical choice. Additional research is necessary to verify if co-inoculation with efficient strains can improve biological N fixation.

**Keywords:** *Glycine max* (L.) Merr.; nitrogen fixation; nitrogen fertilization; *Bradyrhizobium japonicum*; *Azospirillum brasilense*; nodulation; co-inoculation





## 1. Introduction

Soybean [*Glycine max* (L.) Merrill] is grown in a wide variety of soil and climatic conditions and is the dominant oilseed crop in the USA [1]. A unique symbiotic relationship between the soybean plants and the bacteria *Bradyrhizobium japonicum* provides most of the nitrogen (N) required [2]. Biological nitrogen fixation (BNF) is one of the most important environmental and economic processes affecting soybean production; however, as soybean yields have been increasing over the last decades [1], there is a concern whether BNF alone will be able to supply enough N and maintain seed quality at higher yield levels [3,4].

Soybean nodule formation usually starts shortly after emergence but active $N_2$ fixation will only start at the V2 to V3 [5] growth stages. Maximum $N_2$ fixation through BNF occurs between the R3 and R5 growth stages and a reduction is observed between R5 and R7 [6], which potentially limits the N availability during seed filling. Biological N fixation is estimated to provide around 60% of N to soybean plants, the remaining amount will come from the soil and via supplemental fertilization [2,7,8].

The use of plant-growth-promoting rhizobacteria (PGPR) could be an alternative to address the potential soybean N deficit and at the same time maintain a low environmental impact. *Azospirillum brasilense*, a nitrogen-fixing diazotrhoph [9], is a PGPR that in a

symbiotic relationship can stimulate root hair formation and root growth through the production of phytohormone Indole-3-Acetic Acid (IAA), increasing plant absorption of water and nutrients [10–12]. Soybean plants inoculated with strains of *B. japonicum* and *A. brasilense*, which is also called co-inoculation, increased plant growth and BNF when compared to *B. japonicum* alone [13]. Co-inoculation of these two bacteria may lead to an increase in $N_2$ fixation [14] and studies conducted in Brazil have shown similar trends [15–18]. A meta-analysis with *Bradyrhizobium* spp. and *A. brasilense* co-inoculation in Brazil showed significant increases in root mass, nodule number, nodule mass, shoot N content, and grain yield and the response was more prominent in no-till systems, sandy soils, and when grain yields were below 3500 kg ha$^{-1}$ [18].

Another experiment in Brazil compared the co-inoculation (*B. japonicum* and *A. brasilense*) with single inoculation of *B. japonicum,* and also with a non-inoculated treatment with 200 kg N ha$^{-1}$ [16]. The co-inoculated treatment yielded significantly higher than the *B. japonicum* alone across all the environments and was not different from the treatment with N fertilizer [16]. In the USA, a multi-state experiment conducted over 25 site-years showed a positive impact of the co-inoculation on yield only at three sites, and two of them were located in eastern North Dakota [19].

If soil N is inadequate to meet plant needs during early growth when BNF is not occurring yet, a low rate of N fertilizer may be beneficial to stimulate plant growth [20]; however, early season N fertilization can reduce BNF [8]. Supplemental N fertilization has the potential to achieve maximum yields when the N requirement is not met via N supply (BNF and soil) [2]. The BNF activity can be limited by several environmental conditions such as soil moisture, soil compaction, soil pH, temperature, pests, and diseases [21,22]. Late-season application of N has the potential to increase soybean seed protein content [23].

Currently, only one co-inoculation study has included the northern soybean region in the USA and Canada [19]. Additional studies are necessary, as there might be a potential to enhance BNF and sustainably improve soybean productivity. Soybean research on understanding its response to supplemental N compared to only inoculation with *B. japonicum*, as well as being co-inoculated with *A. brasilense*, becomes of great value to soybean producers and has not been reported for the northern USA and southern Canadian soybean growing regions. The objectives of this research were to (1) evaluate soybean response to co-inoculation compared with N fertilization; (2) measure the effect of *Rhizobium* inoculation, co-inoculation, or N fertilization on nodulation. Additionally, a partial profit analysis was conducted to determine the economic return of each treatment.

## 2. Materials and Methods

### 2.1. Experimental Design

The experiment was conducted at three locations in 2021 and three locations in 2022 in the northern USA (Table 1). The sites were located at the North Dakota State University (NDSU) seed farm, Casselton, ND (46.878045°, −97.250898°); NDSU research station at Fargo, ND (46.932124°, −96.858941°); on a soybean producer's field, Lisbon, ND (46.441010°, −97.802389°); and the NDSU research station at Prosper, ND, USA (47.001663°, −97.112775°). The Fargo research station had two separate experiments: one on a naturally drained field (Fargo 1) and the second on a drain-tiled field (Fargo 2), equipped with water table control structures [24]. Therefore, there were four environments each year. Soil samples were taken prior to planting at 0–15 and 15–60 cm for soil characterization and analyzed at NDSU soil testing laboratory (Table 2).

**Table 1.** Previous crop, soil series, and soil taxonomy at Casselton, Fargo, Lisbon, and Prosper, ND, USA in 2021 and 2022.

| Location | Year | Previous Crop | Soil Series [A] | Soil Taxonomy |
|---|---|---|---|---|
| Casselton | 2021 | Wheat [B] | Kindred-Bearden | Fine–silty, mixed, superactive, frigid Typic Endoaquolls<br>Fine–silty, mixed, superactive, frigid Aeric Calciaquolls |
| Fargo (1 and 2) | 2021 and 2022 | Wheat | Fargo-Ryan | Fine, smectitic, frigid Typic Epiaquerts<br>Fine smectitic, frigid Typic Natraquerts |
| Lisbon | 2021 and 2022 | Corn [C] | Barnes-Svea | Fine–loamy, mixed, superactive, frigid Calcic Hapludolls<br>Fine–loamy, mixed, superactive, frigid Pachic Hapludolls |
| Prosper | 2022 | Wheat | Kindred-Bearden | Fine–silty, mixed, superactive, frigid Typic Endoaquolls<br>Fine–silty, mixed, superactive, frigid Aeric Calciaquolls |

[A] Soil data obtained from USDA [25]. [B] *Triticum aestivum* (L.) and [C] *Zea mays* (L.).

**Table 2.** Spring soil test results at all environments in 2021 and 2022.

| Environment | Depth | NO$_3$-N | P | K | SO$_4$-S | pH [A] | OM [B] | EC [C] |
|---|---|---|---|---|---|---|---|---|
| | cm | kg ha$^{-1}$ | mg kg$^{-1}$ | | kg ha$^{-1}$ | | % | mmhos cm$^{-1}$ |
| | | | | **2021** | | | | |
| Casselton | 0–15 | 45 | 16 | 445 | 5 | 7.7 | 4.9 | 0.97 |
| | 15–60 | 47 | 4 | 225 | 72 | 7.9 | 3.6 | 1.58 |
| Fargo 1 | 0–15 | 56 | 31 | 625 | 19 | 7.6 | 5.1 | 0.85 |
| | 15–60 | 111 | 9 | 350 | 165 | 8.1 | 3.6 | 0.95 |
| Fargo 2 | 0–15 | 41 | 32 | 575 | 12 | 7.6 | 5.4 | 0.77 |
| | 15–60 | 101 | 14 | 355 | 30 | 8.0 | 3.7 | 0.85 |
| Lisbon | 0–15 | 28 | 15 | 258 | 19 | 7.5 | 4.2 | 0.61 |
| | 15–60 | 37 | 4 | 150 | 34 | 8.0 | 2.5 | 0.50 |
| | | | | **2022** | | | | |
| Fargo 1 | 0–15 | 10 | 26 | 497 | 16 | 8.0 | 5.4 | 0.40 |
| | 15–60 | 20 | 3 | 425 | 81 | 7.9 | 3.5 | 0.71 |
| Fargo 2 | 0–15 | 10 | 21 | 509 | 11 | 7.8 | 5.5 | 0.49 |
| | 15–60 | 24 | 2 | 385 | 61 | 8.0 | 3.5 | 0.72 |
| Lisbon | 0–15 | 4 | 20 | 285 | 13 | 6.2 | 4.8 | 0.24 |
| | 15–60 | 30 | 4 | 230 | 40 | 7.2 | 2.6 | 0.37 |
| Prosper | 0–15 | 1 | 39 | 241 | 11 | 6.4 | 4.1 | 0.20 |
| | 15–60 | 10 | 4 | 193 | 34 | 7.9 | 2.2 | 0.35 |

[A] pH = in water, [B] OM = organic matter, [C] EC = soil electrical conductivity.

The experiment was a randomized complete block design (RCBD) with four replications per environment and seven treatments (Table 3). The experimental unit was 1.52 × 7.62 m, being four rows wide with 30.5 cm row spacing. The soybean cultivar used for the 2021 and 2022 growing seasons were from Channel (0819R2X, 0.8 maturity) and Asgrow (Fargo and Prosper AG07XF2, 0.7 maturity; Lisbon AG09XF0, 0.9 maturity). Both seed brands are owned by Bayer (Monheim, Germany). The seed was pre-treated by the seed company with Acceleron (i.e., pyraclostrobin and metalaxyl) fungicide seed treatment.

All soybean treatments were inoculated with *Bradyrhizobium japonicum* (America's Best Inoculant, Advanced Biological Marketing, Van Wert, OH, USA) at the rate of 3.2 mL kg $^{-1}$. The treatment we call 'Azospirillum' (Table 3), was co-inoculated. In addition to *Bradyrhizobium japonicum*, this treatment was inoculated with *Azospirillum lipoferum* and *Azospirillum brasilense* (MicroAZ-ST dry; TerraMax Inc., Eagan, MN, USA) at a rate of 2.5 g kg$^{-1}$ of seed.

A range of N rates, applied as urea and ammonium sulfate, were included to produce a response curve. Although soil samples showed S rates were acceptable for soybean, ammonium sulfate was used and balanced all N-rate treatments to have 34 kg S ha$^{-1}$. For the zero N treatment, 34 kg S ha$^{-1}$ was applied as gypsum. Fertilizer treatments were applied by hand on the day of planting and R3 stage to simulate broadcast application. For the split applications, half rate was spread at planting and half rate at the R3 stage to provide N during the reproductive phase.

**Table 3.** Experiment treatments, sources, and stages of application.

| Application Timing | | Total N Applied |
| --- | --- | --- |
| Planting | R3 Growth Stage | |
| N Source | N Source | N Rate |
| kg ha$^{-1}$ | | kg ha$^{-1}$ |
| *B. Japonicum* | - | 0 |
| *Azospirillum* [A] | - | 0 |
| Zero N (0) [B] | - | 0 |
| AMS [C] (30) | - | 30 |
| Urea + AMS (28) | Urea + AMS (28) | 56 |
| Urea + AMS (56) | Urea + AMS (56) | 112 |
| Urea + AMS (168) | Urea + AMS (168) | 336 |

Note: All treatments were inoculated with *Bradyrhizobium japonicum*. [A] In addition to *B. japonicum*, also co-inoculated with *Azospirillum lipoferum* and *A. brasilense*. [B] Between brackets is the amount of N applied at planting and R3. [C] Ammonium sulfate.

All experimental units were monitored and received locally recommended management for control of weeds, pests, and diseases [26]. The experiment was planted as soon as the conditions were favorable in early to mid-May, with a Hege 1000 no-till planter (Hege Company, Waldenberg, Germany) at a seeding rate of 370,000 live seeds ha$^{-1}$. Plant density was determined shortly after soybean emergence (VE) by counting all plants within 1 m of two inner rows of each experimental unit.

Number of nodules and nodule size observations were completed at R2 and R6 stages on *B. japonicum*, co-inoculated (*B. japonicum* with *Azospirillum lipoferum* and *A. brasilense*), and 112 kg N ha$^{-1}$ treatments, at Fargo 1, with four replicates in 2021 and 2022. Samples of 10 plants of each treatment were collected at R2 and R6. The experiment had a non-destructive experimental unit to generate yield data and a destructive unit, adjacent to the yield unit, where plant samples were dug up with a spade, and the roots of each plant were rinsed in a water bucket to remove soil particles. The nodules on each plant were counted and rated for size. The nodule size rating was performed by number of small (<1 mm), medium (1–4 mm), and large (>4 mm) nodules. Volume of the nodules was calculated considering the volume of a sphere ($4/3\pi r^3$), and for the calculation, the small, medium, and large nodules, the diameters of 1, 2.5, and 4 mm were used, respectively. Data were averaged across the 10 observed plants. Dates of field operations and measurements are provided (Table 4).

The experimental units were harvested using a Wintersteiger Classic plot combine (Wintersteiger Ag, Ried, Austria) after the crop reached physiological maturity and harvestable moisture content was reached. Soybean seed samples were cleaned using a Clipper seed cleaner (Ferrell-Ross, Bluffton, IN, USA), and the samples were weighed for yield and thousand-seed weight on a Mettler Toledo XS6001S scale (Mettler-Toledo, LLC, Columbus, OH, USA). A DA 7250 near-infrared (NIR) analyzer measured the oil and protein content (Perten Instruments, Inc., Springfield, IL, USA). The total protein content yield in kg ha$^{-1}$ was calculated by multiplying the seed yield (kg$^{-1}$) with the protein content (g kg$^{-1}$). Moisture and test weight were determined using a GAC 2100 moisture tester (DICKEY-John Corp., Minneapolis, MN, USA), and yield, protein, and oil were corrected to 13% moisture content.

For the economic analysis, the partial profit is (Yield × price) − (N applied × price). Sulfur was not included in the calculation. For the soybean price, the market price of 0.52 USD/kg$^{-1}$ at a Casselton, USA, elevator in mid-April 2023 [27] was used, and the N fertilizer had a price of 1.10 USD/kg$^{-1}$ [28]. The estimated price of 8 USD/ha$^{-1}$ was used for *Azospirullum* (personal communication with supplier). The partial profit was calculated for each experimental unit.

**Table 4.** Dates of field operations and measurements at Casselton, Fargo, Lisbon, and Prosper, ND, USA in 2021 and 2022.

| Operation/Measurement | Casselton | Fargo 1 and 2 | Lisbon | Prosper |
|---|---|---|---|---|
| | | **2021** | | |
| Soil test pre-plant characterization | 6 May | 10 May | 7 May | - |
| Planting/fertilizer application | 6 May | 10 May | 7 May | - |
| Stand count | 15 June | 16 June | 16 June | - |
| Nodule count R2 | - | 20 July | - | - |
| R3 fertilizer application | 15 July | 20 July | 20 July | - |
| Nodule count R6 | - | 30 Aug | - | - |
| Height measurement | 16 Sept | 23 Sept | 21 Sept | - |
| Harvest | 23 Sep | 29 Sep | 28 Sep | - |
| | | **2022** | | |
| Soil test pre-plant characterization | - | 25 May | 24 May | 23 May |
| Planting/fertilizer application | - | 25 May | 24 May | 23 May |
| Stand count | - | 22 June | 21 June | 23 June |
| Nodule count R2 | - | 21 July | - | - |
| R3 fertilizer application | - | 27 July | 26 July | 25 July |
| Nodule count R6 | - | 5 Sept | - | - |
| Height measurement | - | 26 Sept | 23 Sept | 26 Sept |
| Harvest | - | 3 Oct | 5 Oct | 4 Oct |

## 2.2. Weather Conditions

The weather data were collected from the North Dakota Agricultural Weather Network [29]. In 2021, all locations received between 52 and 68% of normal precipitation, and in 2022 all locations received 75 to 87% of normal precipitation (Table 5). In 2021, the average air temperature was 1 °C higher compared to normal at Casselton and Prosper and was 2 °C higher at Fargo and Lisbon. In 2022, the average temperature was 1 °C higher compared to normal at Fargo and Lisbon (Table 5).

**Table 5.** Mean monthly air temperatures and precipitation during growing season at Casselton, Fargo, Lisbon, and Prosper, ND, USA in 2021 and 2022.

| | Average Air Temp | | | Precipitation | | |
|---|---|---|---|---|---|---|
| **Month** | **2021** | **2022** | **Normal [A]** | **2021** | **2022** | **Normal** |
| | | **°C** | | | **mm** | |
| | | | *Casselton and Prosper* | | | |
| May | 13 | 13 | 13 | 20 | 103 | 78 |
| June | 22 | 21 | 19 | 48 | 93 | 109 |
| July | 22 | 22 | 21 | 25 | 85 | 91 |
| Aug | 21 | 20 | 20 | 47 | 56 | 67 |
| Sept | 18 | 16 | 15 | 76 | 24 | 69 |
| Average | 19 | 18 | 18 | Total | 216 | 361 | 413 |
| | | | *Fargo* | | | |
| May | 14 | 13 | 14 | 9 | 87 | 83 |
| June | 23 | 21 | 19 | 89 | 67 | 109 |
| July | 24 | 23 | 22 | 18 | 97 | 84 |
| Aug | 22 | 21 | 20 | 76 | 68 | 66 |
| Sept | 18 | 17 | 16 | 83 | 14 | 69 |
| Average | 20 | 19 | 18 | Total | 276 | 332 | 411 |
| | | | *Lisbon* | | | |
| May | 13 | 13 | 13 | 28 | 105 | 79 |
| June | 22 | 21 | 19 | 108 | 53 | 86 |
| July | 23 | 22 | 21 | 23 | 65 | 73 |
| Aug | 22 | 21 | 20 | 52 | 45 | 67 |
| Sept | 18 | 16 | 15 | 44 | 9 | 65 |
| Average | 20 | 19 | 18 | Total | 255 | 276 | 371 |

[A] Normal represents the 30-year average from 1991 to 2020. Data obtained from North Dakota Agricultural Weather Network [29].

### 2.3. Data Statistical Analysis

Statistical analyses were conducted using a randomized complete block design in which the environment (location experiment) and replication were considered random effects and treatment as fixed effect. Data and dependent variables were analyzed using analysis of variance with statistical software JMP Pro 15 (SAS Institute Inc., Cary, NC, USA). The data were first analyzed for each environment, and after confirming, homogeneity of variances according to Bartlett's Chi-Square test, the data were combined and analyzed across environments. Treatment means were separated using Fisher's protected least significant difference (LSD) at the 95% level of confidence ($\alpha = 0.05$). Regression analyses were performed with the use of JMP Pro 15 (SAS Institute Inc., Cary, NC, USA), where modeling the relationship between two variables was possible.

## 3. Results and Discussion

### 3.1. Plant Population

The plant population was significantly lower at the 30 kg N ha$^{-1}$ and 336 kg N ha$^{-1}$ rates compared to 0 kg N ha$^{-1}$, and the highest N rate significantly lowered the plant population compared to any other treatment (Table 6). A plant population reduction of 5%, with the application of 56 kg N ha$^{-1}$, was reported in a previous study in North Dakota [30]. Ammonia toxicity, resulting in reduced plant population, has been seen in several crops such as corn [31], wheat [32,33], rapeseed (*Brassica napus* L.), and other crops [32]. The reduced plant population in our study could be due to ammonia volatilization toxicity damaging seedlings very early in the season; however, the 56 kg ha$^{-1}$ N and 112 kg N ha$^{-1}$ application rates did not negatively affect the plant population.

**Table 6.** Mean agronomic trait observations for seven treatments averaged across eight environments, in 2021 and 2022.

| N | PP [A] | HT | PC | OC | Yield | SWT | PY | Partial Profit [B] |
|---|---|---|---|---|---|---|---|---|
| kg ha$^{-1}$ | Plants ha$^{-1}$ | cm | g kg$^{-1}$ | g kg$^{-1}$ | kg ha$^{-1}$ | g | kg ha$^{-1}$ | USD/ha$^{-1}$ |
| *B. Japonicum* | 291,398 [a] | 69.6 [b] | 340 [b] | 186.3 [bc] | 3398 [c] | 137.2 [c] | 1158 [b] | 1767 [a] |
| *Azospirillum* [C] | 279,742 [ab] | 69.6 [b] | 338 [bc] | 187.6 [ab] | 3340 [c] | 134.9 [c] | 1132 [b] | 1729 [a] |
| 0 | 285,907 [a] | 70.5 [b] | 339 [bc] | 187.4 [ab] | 3319 [c] | 135.7 [c] | 1127 [b] | 1726 [a] |
| 30 | 265,061 [b] | 71.4 [ab] | 337 [c] | 188.3 [a] | 3410 [c] | 137.1 [c] | 1152 [b] | 1740 [a] |
| 56 | 288,933 [a] | 70.9 [b] | 339 [bc] | 186.3 [bc] | 3472 [bc] | 138.0 [bc] | 1180 [b] | 1744 [a] |
| 112 | 293,080 [a] | 73.1 [a] | 340 [b] | 186.5 [bc] | 3616 [ab] | 141.4 [b] | 1234 [a] | 1757 [a] |
| 336 | 235,248 [c] | 70.8 [b] | 345 [a] | 184.9 [c] | 3684 [a] | 146.1 [a] | 1274 [a] | 1546 [b] |
| ANOVA Significance | ** | * | ** | ** | ** | *** | *** | *** |
| LSD (0.05) | 15,267 | 2.1 | 3 | 1.7 | 159 | 3.7 | 54 | 83 |

Note: *, **, *** = significant at ($p \leq 0.05$), ($p \leq 0.01$), and ($p \leq 0.001$), respectively. Means in a column followed by the same letter are not significantly different at ($p \leq 0.05$). [A] PP = plant population; HT = plant height; PC = protein content; OC = oil content; SWT = thousand-seed weight; PY = protein yield (=Yield × percent protein content). [B] Partial profit (Yield × 0.52 USD/kg$^{-1}$) − (N applied × 1.10 USD/kg$^{-1}$), and *Azospirillum* cost at 8 USD/ha$^{-1}$. [C] Co-inoculation with *Bradyrhizobium japonicum* and *Azospirillum lipoferum* and *A. brasilense*.

### 3.2. Plant Height

Plant height at physiological maturity was different among treatments and the greatest plant height was observed in the treatment group with 112 kg N ha$^{-1}$, averaging 73.1 cm (Table 6). Several researchers, as summarized in a review [34], reported increased plant height with increased N rates of 40 to 80 kg N ha$^{-1}$. In North Dakota, there was a significantly greater plant height with 28 and 56 kg N ha$^{-1}$ [30]. We anticipated greater plant height with an increased N rate; however, that was not the case in this study.

### 3.3. Protein Content

Soybean seed protein content was significantly different among treatments. The treatments' mean protein concentration ranged from 337 to 345 g kg$^{-1}$ (Table 6). Nitrogen increased protein content when 336 kg N ha$^{-1}$ was applied (Table 6).

A significant positive linear relationship between protein content and yield was observed (Figure 1). Similarly, in a soybean production study comparing dry land to irrigated land, soybean yield and protein content were both increased under irrigated fields [35]. A possible explanation is that at the time of partitioning N, there was enough available N and moisture in the soil for plants to fill the seeds and increase protein content; however, most researchers have found different results, where the protein was negatively correlated to yield [36–39], or no relationship between the two traits was observed [40].

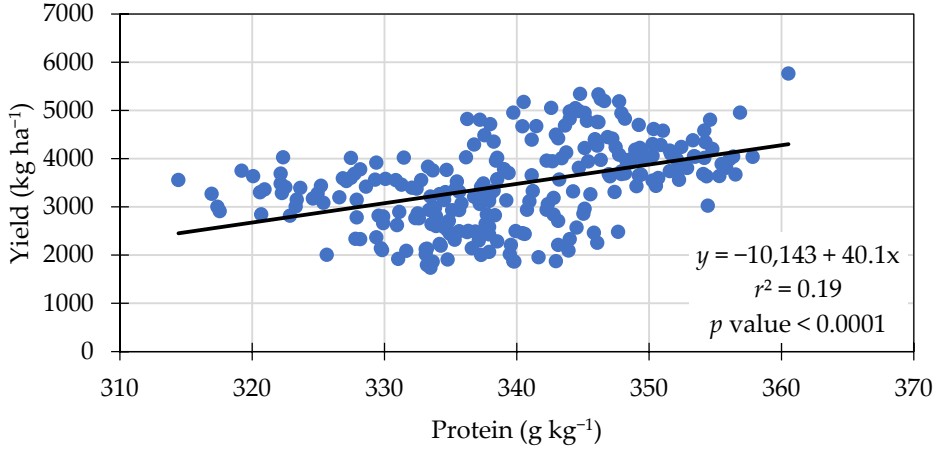

**Figure 1.** Positive linear relationship between seed protein content and seed yield across treatment means in 2021 and 2022.

### 3.4. Oil Content

Seed oil concentration was significantly different among treatments. Treatment means ranged from 184.9 to 188.3 g kg$^{-1}$ in seed oil content (Table 6). The treatments with 30 kg N ha$^{-1}$ resulted in the greatest oil concentration (188.3 g kg$^{-1}$), equal to the *Azospirilum* and 0 kg N ha$^{-1}$. A negative linear relationship was observed between seed protein and oil content (Figure 2), an increase of 1 g kg$^{-1}$ of protein content resulted in a decrease of approximately 0.4 g kg$^{-1}$ in oil content. This negative relationship was also observed by others [37,40]. Another negative linear relationship was observed between oil content and grain yield (Figure 3). The linear equation in Figure 3 indicates yield was approximately 113 kg ha$^{-1}$ lower for each 1 g kg$^{-1}$ oil content increase.

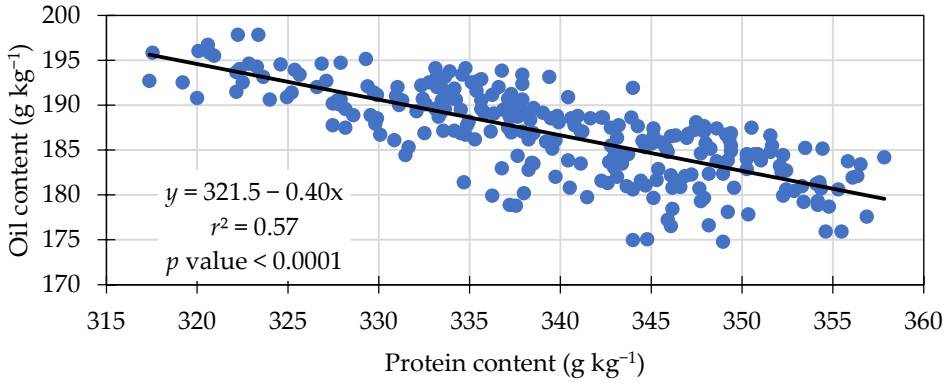

**Figure 2.** Negative linear relationship between seed protein and oil content across treatments in 2021 and 2022.

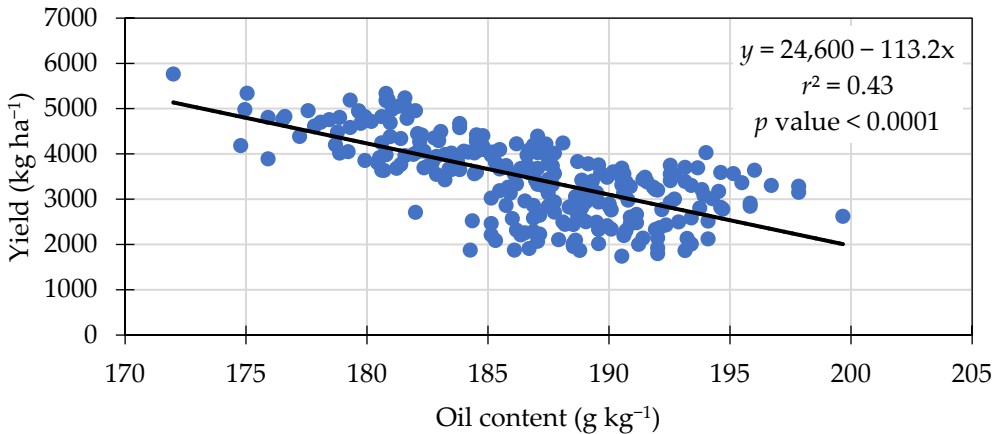

**Figure 3.** Negative linear relationship between seed oil content and grain yield across treatments in 2021 and 2022.

*3.5. Seed Yield*

There were significant grain yield differences among treatments (Table 6). Overall, yield ranged from 3319 to 3684 kg ha$^{-1}$ across the eight environments. The greatest yield was observed when 336 kg N ha$^{-1}$ was applied; however, the established plant population was significantly lower with the application of 336 kg N ha$^{-1}$ (Table 6). The application of 0 kg N ha$^{-1}$ yielded significantly lower compared with the 112 and 336 kg N ha$^{-1}$ yield, with 297 and 365 kg ha$^{-1}$ lower yields, respectively. Lower fertilizer rates of 30 and 56 kg N ha$^{-1}$ did not differ from 0 kg N ha$^{-1}$ (Table 6). The lack of yield response with lower N rates agrees with findings in Manitoba (Canada), where there was no yield response when N rates varied from 0 to 84 kg ha$^{-1}$ [41]. On the other hand, in North Dakota, USA, a significant yield difference was found, with a 5% yield gain using a 56 kg N ha$^{-1}$ N rate compared to no N application [30]. A 5% yield increase with an application of 16 kg N ha$^{-1}$ starter fertilizer was reported in South Dakota, USA [42]. A 2018 study analyzed 207 supplemental N fertilization experiments across the USA with N rates varying from 0 to 560 kg ha$^{-1}$, and showed soybean yield gains of 60 to 120 kg ha$^{-1}$ in N-fertilized over unfertilized treatments [43].

Figure 4 shows a significant polynomial relationship between the N rate and grain yield. The equation suggests that yield increased with the addition of N up to the rate of 273 kg ha$^{-1}$, where the curve hits its highest point (3711 kg ha$^{-1}$). These results of a polynomial relationship are similar to what was found in a corn response to the N rate study, suggesting that corn yield responded to N increments up to a point and then started to decline [44].

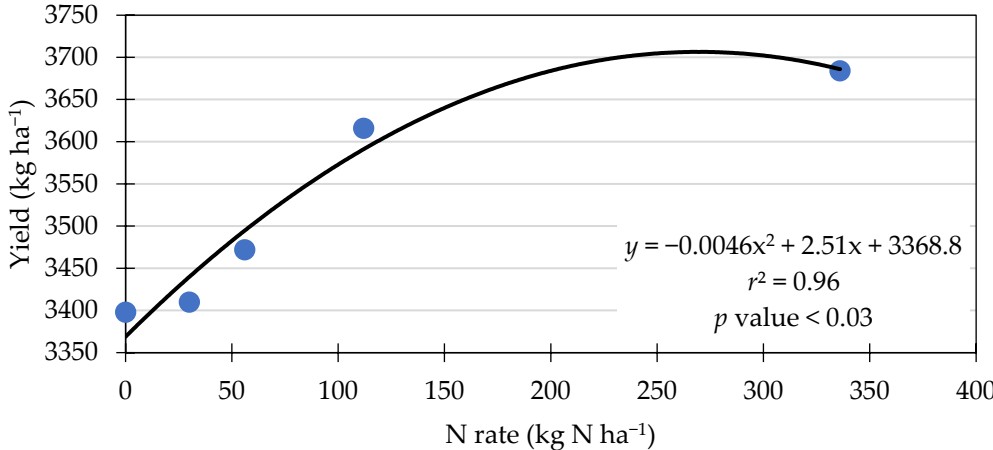

**Figure 4.** Polynomial curve of N rate vs. yield across eight environments in 2021 and 2022.

### 3.6. Thousand-Seed Weight

Soybean thousand-seed weight was significantly different among treatments. Treatments with 112 and 336 kg N ha$^{-1}$ resulted in greater thousand-seed weights than 0 kg N ha$^{-1}$ (Table 6). A significant linear relationship was observed between the thousand-seed weight and yield, in which an increase of 1 g in seed weight resulted in a 33.3 kg ha$^{-1}$ increase in yield (Figure 5).

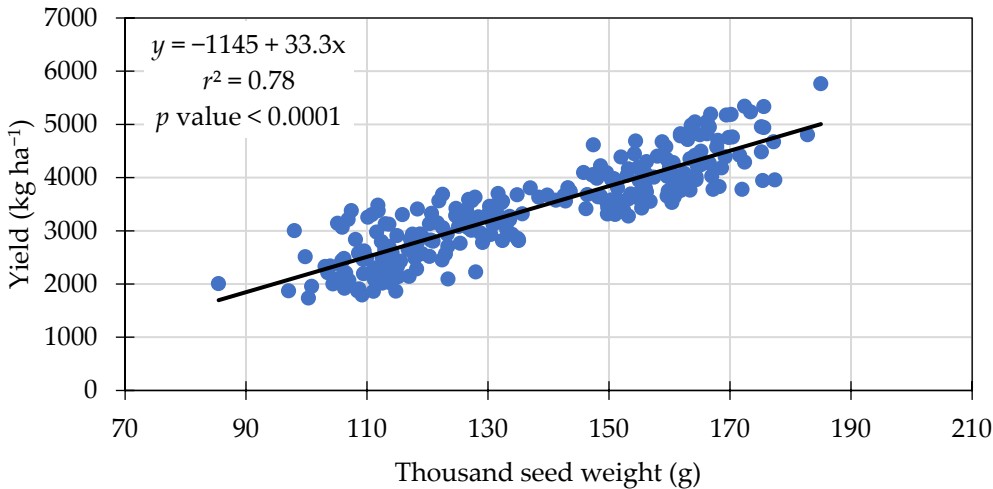

**Figure 5.** Positive linear relationship between thousand-seed weight and grain yield across treatments in 2021 and 2022.

### 3.7. Protein Yield per Hectare

Protein yield was different between treatments. Treatments with 112 and 336 kg N ha$^{-1}$ resulted in greater protein yield ha$^{-1}$ compared to any of the other treatments. (Table 6). In this experiment, we found a positive relationship between yield protein content and yield (Figure 1), providing more total protein ha$^{-1}$ produced for these two N rates.

### 3.8. Partial Profit Analysis

The financial analysis indicates that there were no significant differences in the partial profit except when the highest N rate of 336 kg ha$^{-1}$ was used, which was significantly lower than the other treatments. The treatment with only the application of *B. Japonicum* (1767 USD/ha$^{-1}$) had numerically the highest return. In the analysis, we did not account for the cost of the application of fertilizer and the interest cost if the fertilizer was bought with an operating loan. Therefore, there is no financial incentive to use N fertilization based on our experiment. The economics may change with higher soybean prices and lower N prices.

### 3.9. Co-Inoculation with Azospilillum

There was no significant difference between *Azospirillum* treatment (co-inoculation) and only *B. Japonicum*, for any of the measured traits (Table 6). These results did not agree with a previous study where co-inoculation increased yield by 320 and 690 kg ha$^{-1}$ at Fargo, ND, USA, 2020 and Prosper, ND, USA, 2019, respectively [19]. Despite the significant yield increase in North Dakota reported by De Borja Reis et al. [19], only three out of 25 site-years (including other soybean growing regions in the USA) were responsive to co-inoculation, the authors considered an overall low impact of co-inoculation across the USA and great uncertainty on the response. Most of the studies showing significant yield increases with the use of co-inoculation were conducted in tropical environments. A meta-analysis estimated an average yield gain of 3% with co-inoculation [18]. Environmental factors associated with a positive effect of *Azospirillum* included: no-tillage, sandy soils, low organic matter, drought-prone soils, and lower-yielding environments. Co-inoculation in the furrow at seeding compared with seed-applied inoculant provided a higher seed yield [45]. Due to

variable responses and the low overall number of experiments reported in the literature, additional co-inoculation studies in temperate environments seem necessary for identifying possible responsive environments, inoculant strains, and application methods, before rejecting the potential use of *Azospirillum* in soybean production in the northern growing region of the USA and southern Canada.

### 3.10. Nodulation

Nodulation data from two environments were analyzed separately, and the homogeneity of variances was confirmed; therefore, environments were combined for analysis. There was no effect of treatments on the number of small-size nodules (Table 7); however, the number of medium-size nodules for the *B. Japonicum* treatment at R2 was 18% higher than *Azospirillum* treatment, and 42% higher than 56 kg N ha$^{-1}$. At the R6 stage, the application of N decreased the number of medium size nodules by 30% compared to *B. Japonicum*. *Azospirillum* had the highest number of large nodules at both stages (Table 7), having 2.7 and 1.8 times more large nodules than *B. Japonicum* treatment at R2 and R6 stages, respectively. Nitrogen application had the lowest number of large nodules.

**Table 7.** Treatment effect on average number of nodules per plant classified by small-, medium-, and large-size nodules at R2 and R6 stages, 2021–2022.

| Treatment | Small | | Medium | | Large | | Total | |
|---|---|---|---|---|---|---|---|---|
| | R2 | R6 | R2 | R6 | R2 | R6 | R2 | R6 |
| | ----------------------------Number of nodules---------------------------- | | | | | | | |
| *B. Japonicum* | 6.3 [a] | 4.4 [a] | 24.6 [a] | 36.3 [a] | 1.8 [b] | 3.6 [b] | 32.8 [a] | 44.3 [a] |
| *Azospirillum* [A] | 5.8 [a] | 4.7 [a] | 20.8 [b] | 34.5 [a] | 4.8 [a] | 6.5 [a] | 31.4 [a] | 45.6 [a] |
| 56 or 112 [B] kg N ha$^{-1}$ | 3.8 [a] | 5.4 [a] | 17.3 [c] | 28.0 [b] | 0.1 [c] | 0.8 [c] | 21.2 [b] | 34.2 [b] |
| ANOVA Significance | NS | NS | *** | *** | * | *** | *** | *** |
| LSD (0.05) | - | - | 3.2 | 5.1 | 0.9 | 1.4 | 4.9 | 5.6 |

Note: Means in a column followed by the same letter are not significantly different at ($p \leq 0.05$). *, *** = significant at ($p \leq 0.05$) and ($p \leq 0.001$), respectively. [A] *Azospirillum = B. Japonicum* co-inoculated with *Azospirillum lipoferum + A. Brasilense*. [B] Total of 56 kg N ha$^{-1}$ at planting and 56 kg N ha$^{-1}$ at R3. Total N at R2, 56 kg ha$^{-1}$; and at R6, 112 kg ha$^{-1}$. NS: not significant.

There were no statistical differences in the total number of nodules between the *B. Japonicum* and the *Azospirillum* treatment at R2 or R6; however, the number of nodules with applications of N was significantly lower (Table 7). The volume of nodules differed between treatments for medium- and large-size nodules. For the volume of medium-size nodules at R2, *B. Japonicum* (201 mm$^3$) was significantly higher than co-inoculation, and co-inoculation was significantly higher than the treatment with 56 kg N ha$^{-1}$ (Table 8). At R6, the treatment with 112 kg N ha$^{-1}$ had a lower volume of medium nodules than *B. Japonicum*. The trend for the volume of large nodules was the same at both growth stages, with co-inoculation being the treatment presenting the largest volume of nodules (Table 8).

Due to a higher number of large nodules (Table 7), the co-inoculation was significantly higher in the volume of total nodules at the R2 and R6 observations (Table 8). Nodulation was affected by N application, significantly reducing the number, size, and volume of nodules. This result agrees with other findings, indicating strong inhibition of nodulation when N is applied (high nitrate condition) [46–49]. In another study, reductions of up to 50% in nodule number were observed 15 days after planting—when urea was applied at sowing with rates of 20 to 40 kg N ha$^{-1}$—but these reductions disappeared later on during reproductive stages [50]. In Southern Brazil, a fertilization rate of 200 kg N ha$^{-1}$ decreased nodulation (nodule number, nodule dry weight, and BNF), with no grain yield gain [51]. A significant reduction in the number of nodules at the V4 and R4 soybean growth stages was found, when soybean was treated with 140 and 280 Kg N ha$^{-1}$, accompanied by a higher percent of small nodules as the N rate increased [4].

**Table 8.** Treatment effect on average volume of nodules per plant classified by small-, medium-, and large-size nodules at R2 and R6 stages, 2021–2022.

| Treatment | Small | | Medium | | Large | | Total | |
|---|---|---|---|---|---|---|---|---|
| | **R2** | **R6** | **R2** | **R6** | **R2** | **R6** | **R2** | **R6** |
| | -----------------------------mm$^{-3}$----------------------------- | | | | | | | |
| *B. Japonicum* | 3 [a] | 2 [a] | 201 [a] | 297 [a] | 60 [b] | 119 [b] | 264 [b] | 418 [b] |
| *Azospirillum* [A] | 3 [a] | 2 [a] | 170 [b] | 282 [a] | 161 [a] | 218 [a] | 334 [a] | 502 [a] |
| 56 or 112 [B] kg N ha$^{-1}$ | 2 [a] | 3 [a] | 142 [c] | 229 [b] | 3 [c] | 28 [c] | 147 [c] | 260 [c] |
| ANOVA Significance | NS | NS | *** | *** | *** | *** | *** | *** |
| LSD (0.05) | - | - | 26 | 41 | 30 | 46 | 45 | 64 |

Note: Means in a column followed by the same letter are not significantly different at ($p \leq 0.05$). *** = significant at ($p \leq 0.001$). [A] *Azospirillum* = *B. Japonicum* co-inoculated with *Azospirillum lipoferum* + *A. Brasilense*. [B] Total of 56 kg N ha$^{-1}$ at planting and 56 kg N ha$^{-1}$ at R3. Total N at R2, 56 kg ha$^{-1}$; and at R6, 112 kg ha$^{-1}$. NS: not significant.

In our study, we observed the effects of N inhibiting nodulation, such as a decrease in nodule number and nodule volume; however, the inhibition of nodulation from N cannot be well explained and could be attributed to multiple mechanisms [52,53]. Some of the hypotheses proposed that could explain the nodulation inhibition by N include carbohydrate deprivation in nodules [47]; feedback inhibition by a product of nitrate metabolism [54]; and decreased $O_2$ diffusion into nodules restricting the respiration of bacteroids [55].

A worldwide meta-analysis found that the co-inoculation with *Bradyrizhobium* sp. and *Azospirillum* sp. in multiple studies, from 1987 to 2018, resulted in an average 11% increase in the number of nodules compared to *Bradyrizhobium* alone [56]. Possible explanations for no difference in the number of nodules between *B. Japonicum* and co-inoculated in our study could be attributed to environmental conditions. Other aspects that could have interfered are the bacterial strains, a variable response in different plant genotypes, as well as the quality and quantity of inoculum [56].

Grain yield was also collected for the treatments in the nodulation study, and even though the yield was not different between treatments, the relationship between the volume of nodules and yield was significant in 2021 (Figure 6). The linear equation had an $r^2$ of 0.44, meaning that 44% of the yield could be explained by the volume of nodules. The difference in yield levels between the 2021 and 2022 seasons is evident in Figure 6, and no significant relationship was found in 2022 in the higher-yielding environment. The response in 2021 may have to do with the lower yield potential, as grain yields were below 3500 kg ha$^{-1}$, similar to what was reported by Barbosa [18].

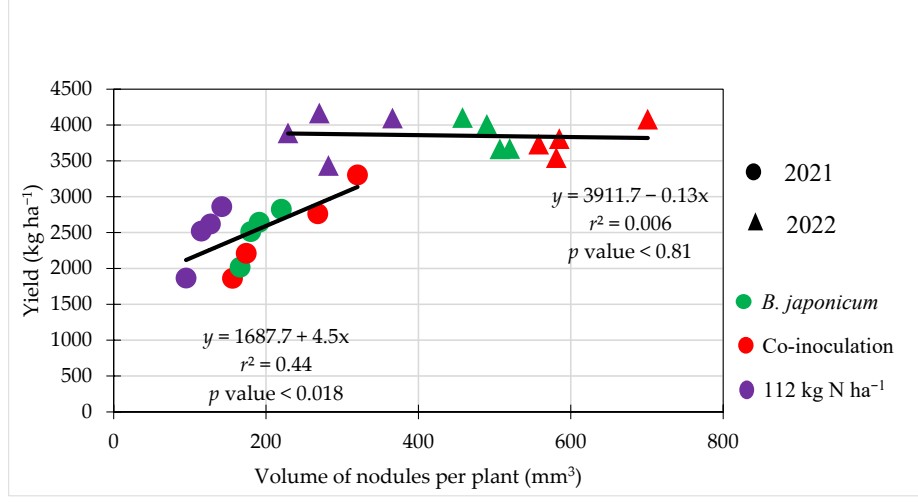

**Figure 6.** Linear relationship between volume of nodules and soybean grain yield at Fargo, ND, USA in 2021 and 2022.

## 4. Conclusions

The higher rates of N (112 and 336 kg N ha$^{-1}$) significantly increased yield, protein content, and thousand-seed weight. Adding N at that highest rate (336 kg N ha$^{-1}$) caused a significant reduction in plant population by 19%. Nitrogen significantly reduced the number and volume of nodules.

When the nodules were classified by size, the number of medium and large nodules was reduced with the application of N. The co-inoculation treatment with *Azospirillum* spp. indicated an apparent nodulation enhancement. Although there were no differences in the total number of nodules, the co-inoculation treatment—when compared with the *B. japonicum* treatment—increased the number of large nodules and, consequently, the total volume of nodules.

Our data did not indicate any yield gain with the use of co-inoculation; however, a significant positive linear regression was observed in the lower-yielding environment between the volume of nodules and grain yield.

Future research exploiting BNF enhancement to produce higher soybean yields would benefit growers, from economic and environmental standpoints. Identifying more responsive environments for using *Azospirillum* spp., using adapted efficient strains for the region, and considering the quantity of inoculum and inoculum placement in furrows compared to on the seed could potentially benefit soybean nodulation and yield. In addition to nodule number and size, future studies should consider including data on nodule BNF efficiency. Yield responses to N fertilizers were modest and variable. Because the yield gain was not enough to offset the cost of fertilizer, northern USA and southern Canada soybean growers should not use supplemental N in soybean.

**Author Contributions:** Conceptualization, J.B. and H.K.; methodology and experimental design J.B., H.K., C.K., E.D. and T.D.; formal analysis J.B. and H.K.; investigation, J.B., H.K. and C.K., writing—original draft preparation, J.B. and H.K.; writing—J.B., H.K., C.K., E.D. and T.D. All authors have read and agreed to the published version of the manuscript.

**Funding:** This research was funded by the North Dakota State University Agricultural Experiment Station, Fargo, ND, USA.

**Data Availability Statement:** The datasets used and/or analyzed in the current study are available from the author upon reasonable request.

**Acknowledgments:** The authors thank Chad Deplazes for assisting with the management of the research, the summer help for their assistance in the project, and the agricultural producer who allowed us to conduct research on his farm.

**Conflicts of Interest:** The authors declare no conflict of interest.

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
