# Peer review of "Soybean Response to N Fertilization Compared with Co-Inoculation of Bradyrhizobium japonicum and Azospirillum brasilense"

_agronomy, doi:10.3390/agronomy13082022_

Round 1
Reviewer 1 Report
Article title:
“Soybean Response to N Fertilization Compared with Co-inoculation of Bradyrhizobium japonicum and Azospirillum brasilense".
The purpose of this study aims to study the effects of Rhizobium inoculation, co-inoculation, or N fertilization on nodulation are measured in soybean response to co-inoculation in comparison to N fertilization, and nodulation. The economic return of each treatment was also determined using a partial profit analysis.
The work done is certainly of international interest and the format applied is certainly suitable for a manuscript. This manuscript dealt with the topic in a different and attractive way, and the titles are related to each other. The work is original, of particular interest, and can certainly stimulate research on this topic.
- Abstract: Add nitrogen fertilization rates and type of fertilizer.
- Keywords: Changing the words so that they are not repeated in the manuscript title.
- Transfer the weather conditions section after 2.1. Section.
- Table 6: Move the units down the parameters.
- Nodulation in results and discussion section: Write the scientific names in italics.
- Nodulation in results and discussion section: 66 or 56 kg N ha-1.
- Conclusion: Rewrite again, highlighting only the most important results.
Author Response
Please, find response below or see attachment
Dear reviewer, I hope this note finds you well. I just wanted to express my gratitude for your feedback
and thoughtful review of this paper. All the comments and observations have truly made a significant
improvement. Thank you once again for your review.
Reviewer 1
- Abstract: Add nitrogen fertilization rates and type of fertilizer.
We inserted the N rates into the abstract. We believe that the type of fertilizer should not be in the abstract because the study was based on N rate and not fertilizer type. However, we spelled out the treatments and fertilizer type in the materials and methods section, on table 3.
- Keywords: Changing the words so that they are not repeated in the manuscript title.
We added “nitrogen fixation” as another keyword
- Transfer the weather conditions section after 2.1. Section.
Sections were transferred.
- Table 6: Move the units down the parameters.
We are not sure what is meant. Currently the units are already bellow the description of each column.
- Nodulation in results and discussion section: Write the scientific names in italics.
Thank you for catching it. Names were corrected to italics.
- Nodulation in results and discussion section: 66 or 56 kg N ha-1.
Thank you for catching this mistake. It is 56 kg N ha-1. Changed.
- Conclusion: Rewrite again, highlighting only the most important results.
We shortened the conclusion with the highlights only.

Reviewer 2 Report
Abstract need to rewritten by incorporating key results of co-inoculation and its effect on yield and amount of N fertilizer can be saved with its application.
In Introductions, Authors may add advanced molecular mechanisms for N fixation and the need for novel strains with higher Nitrogense activities and their relation with plant rhizosphere. Authors should also focus on the novelty of this experiment writing its need and outcome in last paragraph.
In Methodology, authors may add the Molecular accession number, PGPR attributes of the strains, and the repository number from where these strains are available. There is no point of studying a commercial product since the industry must have tested all these parameters before launching this product.
Authors need to study the functional nodule rather a size of the nodule. The leg hemoglobin data need to be added to understand the active nodules.
The discussion needs to be added with recent studies.
Author Response
Please, find the response below or see attachment
Dear reviewer, I hope this note finds you well. I just wanted to express my gratitude for your feedback
and thoughtful review of this paper. All the comments and observations have truly made a significant
improvement. Thank you once again for your review.
Reviewer 2
Abstract need to rewritten by incorporating key results of co-inoculation and its effect on yield and amount of N fertilizer can be saved with its application.
We adjusted the abstract incorporating more co-inoculation results. We added “ However, yield was not different from inoculation with Rizhobium”.
In Introductions, Authors may add advanced molecular mechanisms for N fixation and the need for novel strains with higher Nitrogense activities and their relation with plant rhizosphere.
This paper focused on applied research. We are aware of advancements in molecular mechanisms of N fixation and we are suggesting that future research should focus on utilization of more efficient strains. We added to the abstract “Additional research is necessary to verify if co-inoculation with efficient strains can improve biological N fixation” and deleted “due to greater volume of nodules”
We added “efficient strains” in our conclusions
Authors should also focus on the novelty of this experiment writing its need and outcome in last paragraph.
I believe that we addressed those suggestions in the last paragraph of the introduction section (from lines 73 to 82). Reviewer 3 agreed with us as he wrote:
“Due to variable responses and the low number of experiments reported in the literature, additional co-inoculation studies in temperate environments are necessary for identifying possible responsive environments, inoculant strains, and application methods, for the potential use of Azospirillum in soybean production.”
In Methodology, authors may add the Molecular accession number, PGPR attributes of the strains, and the repository number from where these strains are available. There is no point of studying a commercial product since the industry must have tested all these parameters before launching this product.
Thank you for the comment. Our focus was not on basic research but applied. We reported the product which we use, as this is commercially available. We do recognize that there are several strains, and in the future more efficient strains. The exact product and use rate are reported in a similar way as other relative research, see literature review as example.
Effect of Nitrogen fertilization and inoculation with Bradyrhizobium Japonicum on nodulation and yielding of soybean. Agronomy 2023, 13, 1341. https://doi.org/10.3390/agronomy13051341
Authors need to study the functional nodule rather a size of the nodule. The leg hemoglobin data need to be added to understand the active nodules.
Several papers report on nodule number so it is common in applied research to observe the nodulation. However, you are correct, the functionality of the nodules is critical. We added in the conclusion that future research should also look into nodule efficiency/activity.
The discussion needs to be added with recent studies.
In our discussions in each section we used studies that were current with the research and relevant to our topic.

Reviewer 3 Report
I commend the authors for the analysis methods used, the analysis of the experimental data, the experimental concept and the topic addressed. I propose to the authors to continue and expand the research in time considering the complexity and importance of the field.
Due to variable responses and the low number of experiments reported in the literature, additional co-inoculation studies in temperate environments are necessary for identifying possible responsive environments, inoculant strains, and application methods, for the potential use of Azospirillum in soybean production.
Considering the socio-economic importance of the field, studies should be directed, funded and carried out in the future and validated in experimental field conditions, agricultural agro-funds and different climatic conditions.
Author Response
Please find the response below or see the attachment
Dear reviewer, I hope this note finds you well. I just wanted to express my gratitude for your feedback
and thoughtful review of this paper. All the comments and observations have truly made a significant
improvement. Thank you once again for your review.
Reviewer 3
I commend the authors for the analysis methods used, the analysis of the experimental data, the experimental concept and the topic addressed. I propose to the authors to continue and expand the research in time considering the complexity and importance of the field.
Due to variable responses and the low number of experiments reported in the literature, additional co-inoculation studies in temperate environments are necessary for identifying possible responsive environments, inoculant strains, and application methods, for the potential use of Azospirillum in soybean production.
Considering the socio-economic importance of the field, studies should be directed, funded and carried out in the future and validated in experimental field conditions, agricultural agro-funds and different climatic conditions.
Thank you for your review. We appreciate your feedback.

Reviewer 4 Report
Soybean is one of the most important crops in the world. As a legume plant, it has the ability to coexist with nodule bacteria and biologically fix nitrogen. Since nodule bacteria are not always present in the soil where soybeans are grown, the beneficial effect is obtained by inoculating the seeds with properly selected bacterial strains. However, biological nitrogen fixation does not fully cover the nitrogen nutrient needs of soybeans, especially the currently cultivated high-yielding varieties. Very often, the use of mineral nitrogen allows for higher yields. For environmental and economic reasons, it is important to determine the correct dose of nitrogen and the time of its use. It depends on many factors. Therefore, in my opinion, the research and results presented in the manuscript are interesting and up-to-date. Manuscript is generally well written. Experiment is arranged properly, the experimental material is sufficient. The results are clearly presented. Tables and figures are adequate. In my opinion manuscript requires only minor changes:
1. The authors did not justify why they used such doses of nitrogen. Why did they test 30, 56, 112 and the next one is as much as 336 kg?
2. Table 2 - How was the pH determined. In KCl, H2O, ...?
3. How were the plants taken to count the root nodules? Were they torn out, dug up?
4. line [222] – maybe it should be ‘protein yield’ instead ‘protein content yield’
5. line [223] - please correct the notation of units
6. line [266] [267] – ‘Soybean thousand seed weight…’ ‘…greater thousand seed weight…’
7. line [275] – ‘greater protein content ha-1’ ? maybe better ‘greater protein yield per ha’
8. line [312] – 66 kg or 56? Same for table 7 and line [385]
9. The ‘Conclusion’ is too long. There is no need to repeat what is discussed in the previous chapter. It should be shortened, more synthetic.
Author Response
Please find the response below or see the attachment
Dear reviewer, I hope this note finds you well. I just wanted to express my gratitude for your feedback
and thoughtful review of this paper. All the comments and observations have truly made a significant
improvement. Thank you once again for your review.
Reviewer 4
soybean is one of the most important crops in the world. As a legume plant, it has the ability to coexist with nodule bacteria and biologically fix nitrogen. Since nodule bacteria are not always present in the soil where soybeans are grown, the beneficial effect is obtained by inoculating the seeds with properly selected bacterial strains. However, biological nitrogen fixation does not fully cover the nitrogen nutrient needs of soybeans, especially the currently cultivated high-yielding varieties. Very often, the use of mineral nitrogen allows for higher yields. For environmental and economic reasons, it is important to determine the correct dose of nitrogen and the time of its use. It depends on many factors. Therefore, in my opinion, the research and results presented in the manuscript are interesting and up-to-date. Manuscript is generally well written. Experiment is arranged properly, the experimental material is sufficient. The results are clearly presented. Tables and figures are adequate. In my opinion manuscript requires only minor changes:
- The authors did not justify why they used such doses of nitrogen. Why did they test 30, 56, 112 and the next one is as much as 336 kg?
On line 115 we reported that the idea of the increasing rates were to reproduce a potential response curve. The assumption behind the jump between the rate of 112 and 336 kg N ha-1 was that with this highest rate we would be sure that N would not be limited.
- Table 2 - How was the pH determined. In KCl, H2O, ...?
pH in water. Added to table 2.
- How were the plants taken to count the root nodules? Were they torn out, dug up?
Plants were dug up from the destructive experimental units with a spade, and rinsed in a water bucket to remove excess of soil. We added this to the materials section.
- line [222] – maybe it should be ‘protein yield’ instead ‘protein content yield’
Thank you for the suggestion. We changed to “protein yield” throughout the document.
- line [223] - please correct the notation of units
Corrected
- line [266] [267] – ‘Soybean thousand seed weight…’ ‘…greater thousand seed weight…’
Added “thousand” to corresponding lines.
- line [275] – ‘greater protein content ha-1’ ? maybe better ‘greater protein yield per ha’
Thank you for the suggestion. We adjusted as suggested.
- line [312] – 66 kg or 56? Same for table 7 and line [385]
Thank you for catching this mistake. It is 56 kg N ha-1. We corrected this issue throughout the whole document.
- The ‘Conclusion’ is too long. There is no need to repeat what is discussed in the previous chapter. It should be shortened, more synthetic.
We shortened the conclusion, and reworded some of it.
